# Differential Recovery Patterns of the Maxilla and Mandible after Eliminating Nasal Obstruction in Growing Rats

**DOI:** 10.3390/jcm11247359

**Published:** 2022-12-11

**Authors:** Mirei Keitoku, Ikuo Yonemitsu, Yuhei Ikeda, Huan Tang, Takashi Ono

**Affiliations:** Department of Orthodontic Science, Graduate School of Medical and Dental Sciences, Tokyo Medical and Dental University (TMDU), Tokyo 113-8510, Japan

**Keywords:** nasal obstruction, recovery, optimal timing, maxillofacial growth, mouth breathing

## Abstract

Although nasal obstruction (NO) during growth causes maxillofacial growth suppression, it remains unclear whether eliminating the NO affects maxillary and mandibular growth differentially. We aimed to clarify whether eliminating NO can help regain normal maxillofacial growth and to determine the optimal intervention timing. Forty-two 4-week-old male Wistar rats were randomly divided into six groups. Their left nostril was sutured to simulate NO over different durations in the experimental groups; the sutures were later removed to resume nasal breathing. Maxillofacial morphology was assessed using microcomputed tomography. Immunohistochemical changes in hypoxia-inducible factor (HIF)-1α, osteoprotegerin (OPG), and receptor activator of nuclear factor kappa-B ligand (RANKL) of the condylar cartilage were evaluated to reveal the underlying mechanisms of these changes. Maxillary length was significantly lower in rats with NO for ≥5 weeks. In groups with NO for ≥7 weeks, the posterior mandibular length, ramus height, thickness of the hypertrophic cell layer in the condylar cartilage, HIF-1α levels, and RANKL levels were significantly lower and OPG levels and RANKL/OPG were significantly higher than those in the control group. Our findings suggest that eliminating NO is effective in regaining maxillofacial growth. Moreover, the optimal timing of intervention differed between the maxilla and mandible.

## 1. Introduction

Nasal obstruction (NO) induces mouth breathing and decreases percutaneous oxygen saturation (SpO_2_). The hypoxic condition adversely affects the entire body. Clinical studies have shown that chronic NO induces a variety of symptoms, such as headache, fatigue, sleep disturbances, daytime drowsiness, and distraction, causing the quality of life (QoL) to be abrogated [1,2]. Furthermore, NO affects the hippocampus, leading to memory and learning deficits [3], and also affects taste cells, leading to impaired sweet taste perception [4]. In the orofacial region, mouth breathing is known to cause malocclusion. Several clinical studies have shown that mouth breathing due to NO propels downward migration of the palatal plane [5], narrowing of the maxilla, elevation of the palate [6], and vertical growth of the mandible [7]. A clinical study showed that nasal congestion leads to functional modulation of the neuromuscular system that alters craniofacial bone structure, tongue and mandible position, and soft tissues, causing “adenoid face” features [8]. These findings indicate that NO causes deterioration of growth, particularly in the maxillofacial region.

One of the factors that change the maxillofacial configuration due to NO is the physiological effect of the muscles. NO has been shown to inhibit masticatory muscle formation and contractile properties and suppress bone growth at muscle attachment sites [9]. Another factor is an increase in the hypoxia-inducible factor (HIF) in the mandibular cartilage [10]. HIF is a transcription factor that plays a pivotal role in the adaptive response of cells to hypoxic stress, which increases when systemic SpO_2_ decreases due to hypoxia [11]. HIF-1α promotes osteoblast and osteoclast differentiation, regulates chondrocyte apoptosis [12], and exerts osteoclast activation mechanisms [13]. We previously found increased HIF-1α levels in the condyles of rats with NO, which promotes degeneration of the condylar cartilage and inhibits growth of the condyle [10]. In addition, the activation of osteoclasts occurs in conjunction with an increase in the ratio of osteoprotegerin (OPG), which is involved in osteoclast activation, and receptor activator of nuclear factor kappa-B ligand (RANKL), which suppresses osteoclast activity. Although NO has been reported to significantly affect the growth of the maxillofacial region through several proposed mechanisms, few studies have investigated the aftereffects of eliminating NO.

With regard to the effect of obstruction of the upper airway (UA), maxillofacial growth following treatment of hypertrophied adenoids has been investigated [14,15]. Adenoids are located in the posterior part of the nose, and their enlargement narrows the air passage and interferes with nasal breathing. Adenoids are often enlarged between the ages of 3–6 years, and adenoidectomy is performed in this age group in severe cases when complications, such as NO, sleep-disordered breathing, recurrent acute otitis media, and chronic rhinosinusitis, are present [16,17]. It has been reported that after adenoidectomy and recovery of normal nasal breathing, the reduced mandibular growth is eventually restored to the level in the control group [18], and labial tilting of the upper and lower incisors is improved [19]. Although adenoidectomy is effective between the ages of 4–7 years [16], it is necessary to clarify the maximum age at which intervention for NO can be expected to improve volumetric craniofacial growth.

In humans, maxillary growth peaks at around 10–11 years, while mandibular growth peaks at 12–14 years old [20,21]. Owing to the differential growth between the maxilla and mandible [22], the maxillary growth spurt occurs earlier than that of the mandible. Therefore, the critical periods of therapeutic intervention may differ between the maxilla and mandible. Although improvement in nasal breathing was reported to restore the maxillary growth after elimination of NO in 9-week-old rats [23], few comparative studies have clarified the morphological and histological effects on the maxillary and mandibular bones and investigated the optimal timing of eliminating NO. Therefore, the purpose of this study was to clarify whether the maxillofacial morphology could attain normal growth after elimination of NO and to determine the optimal timing of intervention for NO using an experimental model in growing rats.

## 2. Materials and Methods

### 2.1. Experimental Animal Model

The study protocol was approved by the Institutional Animal Care and Use Committee of Tokyo Medical and Dental University (TMDU) (Approval No. A2019-004C), and experimental procedures were performed in accordance with the TMDU Animal Care Standards and ARRIVE guidelines. 

Forty-two 4-week-old male Wistar rats were randomly divided into 6 groups (*n* = 7 each), as shown in Figure 1. The experimental schedule was as follows: control group, 1-week recovery group (hereafter, W1 group), W3 group, W5 group, W7 group, and W9 group according to differences in the timing of NO elimination (Figure 1a). The left side of the nose of the rats in W1, W3, W5, W7, and W9 groups was sutured with a silk thread under isoflurane inhalation anesthesia (Figure 1b), and NO was eliminated by suture removal in each group every 2 weeks. All rats were euthanized after 13 weeks using CO_2_ gas. The body weight of the rats and SpO_2_ were measured every week at 10:00 a.m. throughout the experimental period. To avoid fluctuations in SpO_2_ caused by body movement, a mouse pulse oximeter (Mouse OX; STARR Life Sciences, Oakmont, PA, USA) was used while administering 4% isoflurane (inhalation anesthetic) (SFMBX1; DS Pharma Biomedical, Osaka, Japan). The heads of the rats were fixed in 4% paraformaldehyde in 0.1 M phosphate buffer, pH 7.4, for 24 h at 4 °C.

### 2.2. Morphological Evaluation of Maxillofacial Bones 

Maxillofacial morphological changes were measured using microcomputed tomography (micro-CT; SMX-100CT; Shimadzu, Kyoto, Japan). Scans were obtained with a source voltage of 75 kV and a source current of 30 mA. Various parts of the maxilla and mandibular bone were measured using a three-dimensional image analysis software program (TRI/3D-BON; Ratoc System Engineering, Tokyo, Japan). The three-dimensional coordinate positions of the biologically relevant cranial landmarks and the line connecting them are defined (Figure 2a,b; Table 1 and Table 2). The air-occupied part was extracted from the CT image (Figure 2c) using 3D-BON, as shown in Figure 2d, and the UA volume was measured. 

### 2.3. Tissue Preparation 

After morphological evaluation by micro-CT, the temporomandibular joints of both sides and their surrounding tissues were decalcified with Osteosoft (Merck Millipore, Burlington, MA, USA) for 3 weeks. They were then embedded in paraffin and cut sagittally into 6 μm thick sections with a microtome.

### 2.4. Histomorphometry with Toluidine Blue Staining

Sections from the center of the condyles were selected and stained with toluidine blue to measure the width of the mandibular condylar cartilage layers and observe the chondrocytes. From the articular surface down, the cartilage of the condyle was divided into four histological layers: fibrous, proliferating cell, mature cell, and hypertrophic cell layers, based on a previous study [24]. The thickness of the four layers in the superior region was measured using image analysis software (NIS-Elements Analysis D, National Institutes of Health, Bethesda, MD, USA) (*n* = 7 each group).

### 2.5. Immunohistochemistry for HIF-1α, OPG, and RANKL 

Immunostaining for HIF-1α, OPG, and RANKL was performed using mandibular cartilage samples of the rats (*n* = 7 each group). The site and thickness of the cartilage were the same as those used for toluidine staining. Deparaffinized sections were pressurized with an antigen activator (Histro VT One; Nacalai Tesque, Kyoto, Japan) at a high temperature for 20 min. Endogenous peroxidase was removed by treatment with a hydrogen peroxide blocking reagent. After washing, the specimens were incubated overnight at 4 °C with the following primary antibodies: monoclonal mouse anti-HIF-1α (1:300, Gene Tex, GTX628480, clone GT10211, Irvine, TX, USA), monoclonal mouse anti-OPG (1:300, Santa Cruz Biotechnology, SC-390518, Dallas, TX, USA), and monoclonal mouse anti-RANKL (1:300, Santa Cruz Biotechnology, SC-377079, Dallas, TX, USA). The slides were then incubated with a secondary antibody (Vectastain ABC Mouse IgG Kit, Vector, PK-4002, Newark, NJ, USA). In each section, the number of HIF-1α-, OPG-, and RANKL-positive cells was counted at least three times within the region of interest (ROI; 150 μm × 200 μm) and averaged. The ROI was selected to cover the total thickness of the proliferative and hypertrophic layers in the superior region of the mandibular condyle. The RANKL/OPG ratio was also calculated to reveal the changes in osteoclastogenesis.

### 2.6. Statistical Analysis 

Results are expressed as the mean ± standard deviation (SD). The data were analyzed using Tukey’s test after testing for normality and equal variances. Moreover, correlations were determined between UA volume and its affected area and between mandibular height and thickness of the hypertrophic cell layer using Pearson’s correlation. In all analyses, statistical significance was set at *p* < 0.05.

## 3. Results

### 3.1. Systemic Changes

There were no significant differences in body weight change between the control and any of the experimental groups of rats throughout the experiment (Appendix A). In the control group, the SpO_2_ remained almost stable throughout the experimental period. SpO_2_ levels were significantly lower in the W1, W3, and W5 groups during the NO period than those in the control group. In contrast, in groups W7 and W9, SpO_2_ significantly decreased throughout the experimental period, regardless of the NO period (Figure 3).

### 3.2. Measurement of the Maxillofacial and UA Morphology

Figure 4 shows the changes in maxillofacial morphology with respect to reopening of the nasal airway during the experimental period. The anterior-posterior length of the maxilla was significantly lower in the W5, W7, and W9 groups than in the control group (Figure 4a). The UA volume was significantly lower in the W1, W3, W5, W7, and W9 groups than in the control group. The UA volume indicated difficulty in catching up with the normal growth pattern even after recovery of the nasal passage. The posterior mandibular length and ramus height were significantly lower in the W7 and W9 groups than in the control group (Figure 4b). The UA volume was significantly correlated with the maxillary length and posterior mandibular length (Figure 5), whereas there was no significant correlation between the total mandibular length and anterior mandibular length.

### 3.3. Histomorphometry with Toluidine Blue Staining

In all groups, the fibrous layer, proliferating cell layer, mature cell layer, and hypertrophic cell layer of condylar cartilages were clearly observed by toluidine blue staining (Figure 6). However, there was a difference in the thickness of each layer, with the hypertrophic cell layer being the thickest. The thickness of the hypertrophic cell layer significantly decreased when the timing of recovery to nasal breathing was delayed (Figure 7). In particular, a significant decrease was observed in the W7 and W9 groups. However, there was no significant change in the thicknesses of the other layers. The thickness of the hypertrophic cell layer showed a significant positive correlation with ramus height (Figure 8). 

### 3.4. Expression of HIF-1α, OPG, and RANKL Protein in the Articular Cartilage

HIF-1α was specifically expressed in hypertrophic cell layers (Figure 9). This was observed in both experimental and control groups. In contrast, OPG and RANKL were specifically expressed at the border between the hypertrophic cell layer and subchondral bone (Figure 10 and Figure 11), which was found in both control and experimental groups. The percentage of HIF-1α-positive cells increased as the NO exposure period increased. It was significantly higher in the W5, W7, and W9 groups than that in the control group (Figure 12). OPG was highly expressed in the control group; however, it showed a significant decrease when the NO persisted for 7 and 9 weeks. Conversely, RANKL was only slightly expressed in the control group and showed a significant increase as the NO period increased from 7 to 9 weeks. The RANKL/OPG ratio was significantly higher in the W7 and W9 groups than in the control group (Figure 12). The percentage of HIF-1α-positive cells showed a significant negative correlation with the thickness of the hypertrophic cell layer (Appendix A). Moreover, SpO_2_ showed significant negative correlations with the percentage of HIF-1α and RANKL-positive cells, and with the RANKL/OPG ratio. In contrast, SpO_2_ showed a significant positive correlation with the percentage of OPG-positive cells (Appendix A).

## 4. Discussion

In this study, we focused on whether it was possible to regain maxillofacial growth by recovery of nasal breathing in growing rats and to determine the optimal timing of intervention for NO based on the growth patterns of the maxilla and mandible. The maxillary length and UA volume were significantly lower in rats with NO for ≥5 weeks and ≥1 week, respectively, as compared with those in the control rats. On the other hand, in the groups with NO for 7 weeks, the posterior mandibular length, ramus height, thickness of the hypertrophic cell layer in the condylar cartilage, HIF-1α levels, and RANKL levels were significantly lower and OPG levels were higher than those in the control group. These different time courses indicate that the maxilla requires earlier elimination of NO to catch up with the normal growth pattern than does the mandible. 

Previous studies have investigated the effects of nasal breathing disorders by blocking only one nostril to decrease SpO_2_ and induce hypoxia [3,4,5,6,7,8,10,25,26]. It has also been found that unilateral NO has little or no effect on the whole body of rats [8,10,27,28]. Similarly, there was no significant difference in body weight between the control and experimental groups in this study, indicating that there was no significant effect on whole-body growth. To observe the effect of eliminating NO, we developed a transient NO model by first suturing the nose with a thread and then inducing recovery by removing the thread, instead of the conventional method of burning one side of the nostril. In fact, during the period of NO, the SpO_2_ of the experimental group was significantly lower than that of the control group. Thus, the transient NO model seems to be appropriate for investigating the effect of recovery from unilateral NO on craniofacial growth.

As for morphological changes in the NO model, the UA volume was significantly smaller in all experimental groups compared with that in the control group. This indicates that it is difficult for the UA to regain its physiological volume even after a short period of NO. Furthermore, the UA volume appeared to have decreased in proportion with the duration of NO. Previous studies have shown that the volume of space inside the nasomaxillary complex varies with the amount of airflow into the nasal cavity [29,30] and that increased external force on the central palatal suture cartilage promotes chondrogenic differentiation [31]. Thus, the findings may be attributable to an insufficient increase in the depth of the nasal cavity and contraction of the maxilla due to the reduced airflow caused by NO, regardless of the duration of SpO_2_ reduction. With regard to maxillary morphological changes, the decrease in the anterior–posterior dimension was the most obvious finding in the three dimensions, with significant reduction in the W5, W7, and W9 groups. A previous study showed that the lateral growth of the maxilla is nearly complete by 4 weeks of age in rats, while the anteroposterior growth of the maxilla is not completed by 4 weeks of age and continues to increase slowly until 10 weeks [32]. In this study, a significant decrease in the anterior–posterior dimension was observed in the W5 group, which comprised 9-week-old rats. This suggests that presence of NO at the age of 9 weeks inhibits anterior–posterior maxillary growth, which is consistent with the age of maxillary growth in rats shown in a previous study [33].

On the other hand, morphometric measurements of the mandible showed that the posterior mandibular length and ramus height were significantly decreased in the W7 and W9 groups. There was also a correlation between the posterior mandibular length and UA volume. Thus, the reduction in UA volume is thought to be a factor for mandibular morphological changes. Other causes of mandibular morphological changes due to NO have been reported extensively [9,10,33] in relation to muscles and SpO_2_. One study suggested that NO causes physiological and functional changes in the muscles attached to the mandible, resulting in mandibular muscular hypoplasia, which is particularly significant in the superficial layer of the masseter muscle [9]. Previous studies have revealed that mandibular muscle and bony growth of the mandible influence each other [34,35,36]. Muscle hypoplasia regulates bone growth during muscle attachment [37]. In our study, the posterior mandible, which was significantly reduced in length in groups W7 and W9, encompassed the vicinity of the gonion. Since the masseter muscle inserts at the site near the gonion, it is inferred that the amount of bone growth at the attachment site is reduced. 

In groups W7 and W9, SpO_2_ levels decreased significantly throughout the experimental period, regardless of the NO period. A significant reduction in posterior mandibular length was also observed in groups W7 and W9. This suggests that the SpO_2_ level may be involved in mandibular morphological changes, as previously suggested [10]. The timing of the decrease in SpO_2_ levels also coincided with a significant decrease in the condylar cartilage thickness, especially in the hypertrophic cell layer, as shown by toluidine blue staining. The hypertrophic cell layer plays a particularly important role in the progressive stages of osteogenesis [38]. In this study, a significant correlation was observed between the thickness of the hypertrophic cell layer and ramus height. If the physiological process of hypertrophy is jeopardized by any factor, the next critical step is osteogenesis [39]. Thus, maintenance of the SpO_2_ level could contribute to the development of hypertrophy [40]. HIF-1α, which is induced by hypoxia [11], was found to be localized in the hypertrophic cell layer. This is consistent with the findings of a previous study [41]. This study also revealed a significant negative correlation between the percentage of HIF-1α-positive cells and the thickness of the hypertrophic cell layer, suggesting that HIF-1α has a significant influence on the growth of the condylar cartilage.

RANKL is expressed by osteoblasts and promotes osteoclast differentiation and bone resorption [42], and OPG is a de novo receptor for RANKL that binds to RANKL and inhibits bone resorption [43]. The RANKL/OPG ratio, calculated from RANKL and OPG expression, is a useful index for predicting the osteoclast environment [44]. In the present study, there was a significant increase in the HIF-1α and RANKL levels and significant decrease in OPG levels occurred significantly in the W7 and W9 groups. Previous studies have shown that increased HIF-1α levels promote an increase in RANKL production and a decrease in OPG production, osteoclast differentiation, and bone resorption [45,46]. In the present study, the timing of the changes in these three factors coincided with the timing of changes in mandibular growth, suggesting that they are associated with bone formation. The percentage of positive cells of HIF-1α, RANKL, OPG, and RANKL/OPG ratio were all found to be significantly correlated (HIF-1α, negative; RANKL, negative; OPG, positive; and RANKL/OPG ratio, negative) with systemic SpO_2_. This suggests that systemic hypoxia affects the local area, the condylar cartilage, and it may be inferred that the osteogenic and osteoclastic systems of the mandible operate sensitively to systemic oxygen deprivation.

There are several limitations to this study. First, rats cannot breathe through their mouths. Because their bodily structures differ from those of humans, the findings of our study cannot be completely extrapolated to humans. Second, the effects of NO before 4 weeks of age were not evaluated. Therefore, it is necessary to develop methods to simulate NO in rats less than 4 weeks old. The rats used should be as young as possible if the effects of growth are to be examined. Finally, the sample size of this study was small. Because we wanted to elucidate the timing of recovery, the increased number of groups forced us to minimize the sample size for each group. Nevertheless, this study suggests that to enable the maxilla and mandible recover the normal growth pattern, the NO should be eliminated by 7 weeks of age for the maxilla and by 9 weeks of age for the mandible. A previous study showed that maxillary growth precedes mandibular growth in rats [47], and this difference in the growth peak may have caused the difference in the recovery time in this study. Likewise, maxillary growth also precedes mandibular growth in humans [48], suggesting that early recovery from NO is necessary for maxillary bone growth.

## 5. Conclusions

Recovery from NO in growing rats restored morphological growth in the anterior-posterior dimension of the maxilla and posterior mandibular length. Our findings also suggest that the optimal timing for intervention was before 7 weeks of age for the maxilla and before 9 weeks of age for the mandible in growing rats. In humans, the growth of the maxilla occurs before that of the mandible same as that in rats. This suggests that early intervention for the maxilla is necessary. The increase in HIF-1 and RANKL levels and decreased in OPG levels with increasing NO duration suggests that NO may affect the osteoclast environment of the mandibular cartilage. Similar changes in the osteoclast environment may occur in humans, and further research is necessary for confirmation.

## Figures and Tables

**Figure 1 jcm-11-07359-f001:**
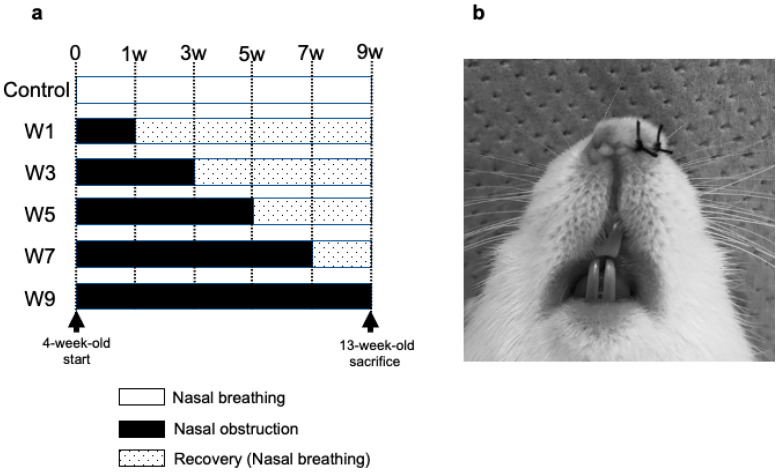
Time schedule of the experiment and implementation of NO. (**a**) Forty-two 4-week-old male Wistar rats were divided randomly into 6 groups (control group, 1-week recovery group (hereafter referred to as W1 group), W3 group, W5 group, W7 group, and W9 group (*n* = 7)). At the time of recovery in the schedule (W1, W3, W5, and W7 groups), the nasal sutures were removed, and the nasal obstruction was eliminated. In the W9 group, the nasal obstruction was maintained throughout the experimental period. All rats were sacrificed at 13 weeks of age; (**b**) the left side of the nostril of 35 rats in the W1, W3, W5, W7, and W9 groups were sutured with two stitches of silk thread.

**Figure 2 jcm-11-07359-f002:**
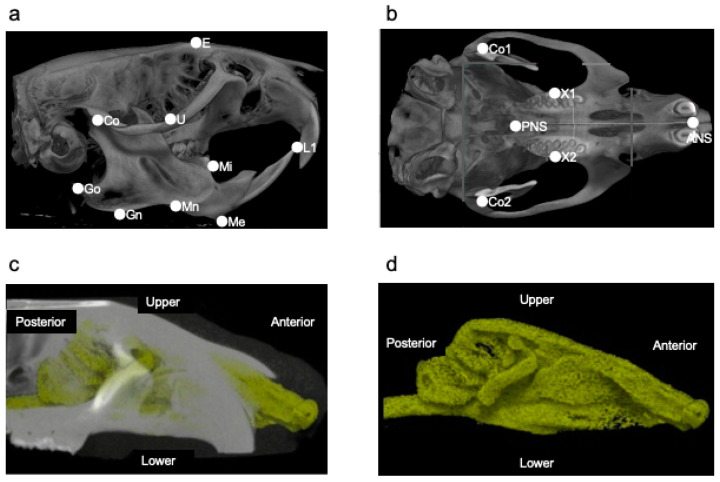
The landmarks used for measurements. (**a**) Landmarks are shown in lateral view. Abbreviations: E—the intersection between the frontal bone and the most superior and anterior point of the ethmoid; U—the intersection between the maxillary sinus and the distal surface of the third superior molar tooth; Co—the most posterior and superior point on the mandibular condyle; Go—the most posterior point on the mandibular ramus; Mn—the most concave portion of the concavity on the inferior border of the mandibular corpus; Gn—the most inferior point on the ramus that lies on a perpendicular bisector of the line Go–Mn; Me—the most inferior and anterior point of the lower border of the mandible; L1—the most anterior and superior point on the alveolar bone of the mandibular incisor; Mi—the junction of the alveolar bone and the mesial surface of the first mandibular molar. (**b**) Landmarks are shown in vertical view. Abbreviations: ANS—the most anterior part of the palate; PNS—the most posterior part of the palate; X1—the most anterior and superior point in the molar process of the right maxilla; X2—the most anterior and superior point in the molar process of the left maxilla. (**c**) The volume of the nasal cavity was measured by extracting the air part (yellow part) from the computed tomography image. (**d**) The air-occupied part extracted from (**c**).

**Figure 3 jcm-11-07359-f003:**
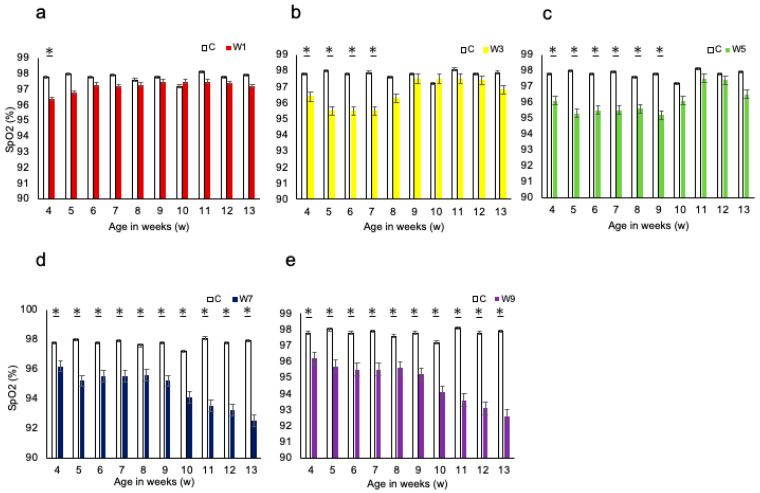
Changes in body weight and SpO_2_. (**a**) Comparison of SpO_2_ between the W1 and control groups throughout the experimental period, *: *p* < 0.05; (**b**) comparison of SpO_2_ between the W3 and control groups throughout the experimental period, *: *p* < 0.05; (**c**) comparison of SpO_2_ between the W5 and control groups throughout the experimental period, *: *p* < 0.05; (**d**) comparison of SpO_2_ between the W7 and control groups throughout the experimental period, *: *p* < 0.05; (**e**) comparison of SpO_2_ between the W9 and control groups throughout the experimental period, *: *p* < 0.05. Abbreviation: SpO_2_—oxygen saturation.

**Figure 4 jcm-11-07359-f004:**
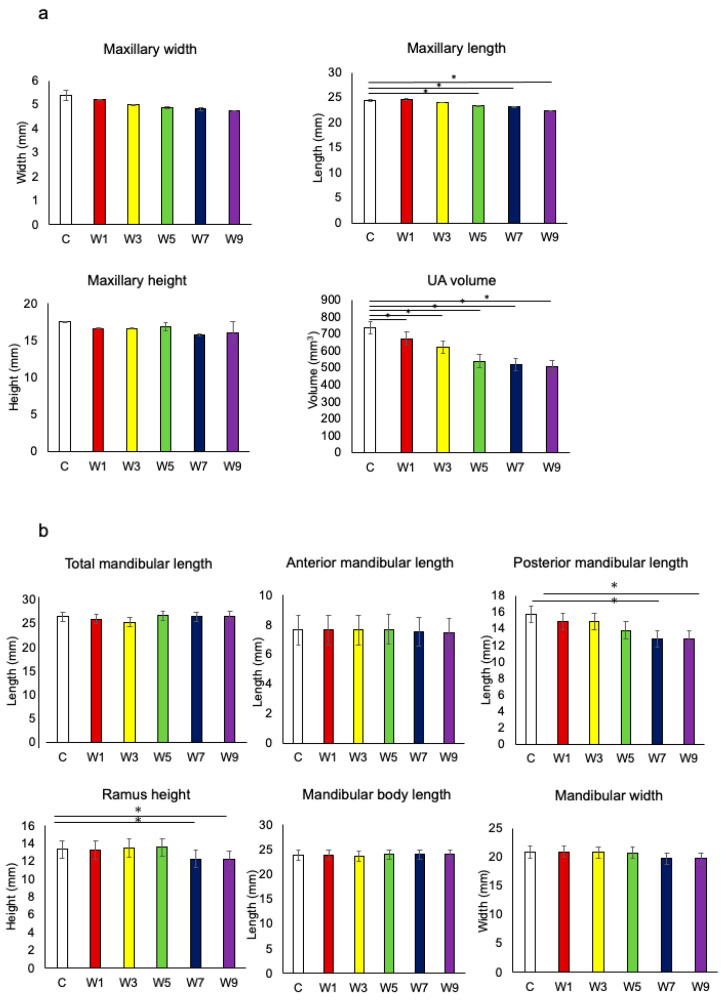
Measurement of the maxillofacial morphology. (**a**) Comparison of the maxillary size in three dimensions and upper airway volume, *: *p* < 0.05. Abbreviation: UA—upper airway; (**b**) comparison of the mandibular size in six dimensions, *: *p* < 0.05.

**Figure 5 jcm-11-07359-f005:**
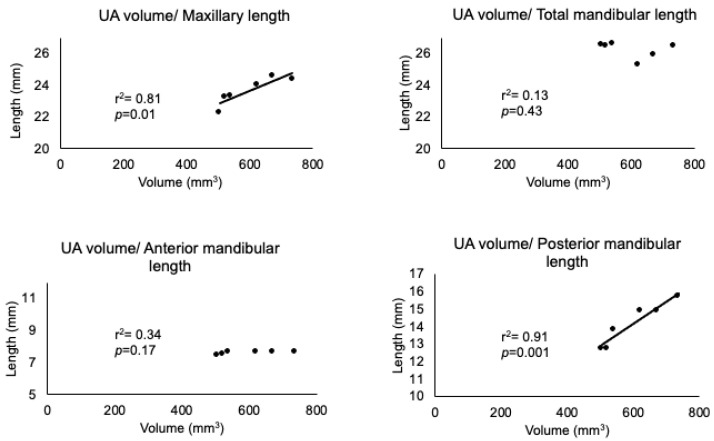
Correlations between upper airway volume and maxillomandibular measurements. Correlations between UA volume and variables in the maxillomandibular skeleton were determined. Abbreviations: UA—upper airway; r^2^—squared correlation coefficient; *p*—probability.

**Figure 6 jcm-11-07359-f006:**
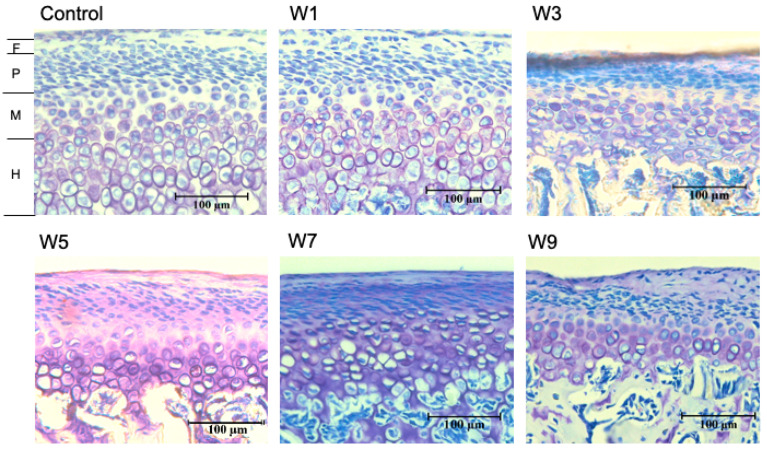
Histomorphometry with toluidine blue staining. Histological staining with toluidine blue in each of the 6 groups (C, W1, W3, W5, W7 and W9). The thickness of the four layers in the superior regions were measured using image analysis software. Abbreviations: F, fibrous layer; P, proliferating cell layer; M, mature cell layer; H, hypertrophic cell layer. Bar depicts 100 μm.

**Figure 7 jcm-11-07359-f007:**
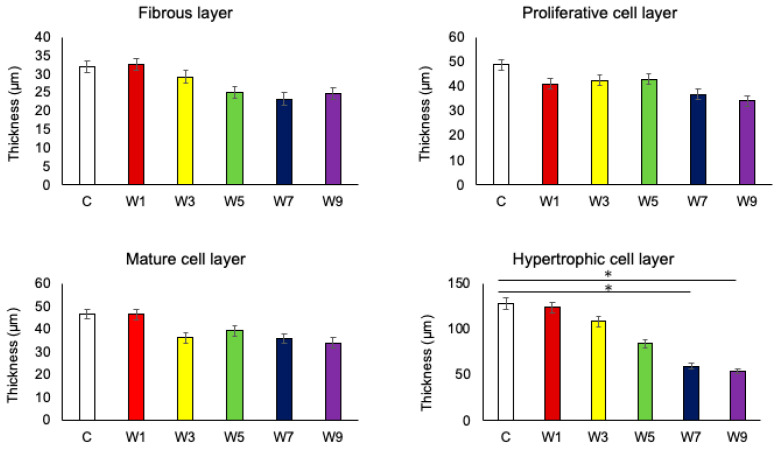
Comparison of the thickness of the four layers of mandibular cartilage in the six groups, *: *p* < 0.05.

**Figure 8 jcm-11-07359-f008:**
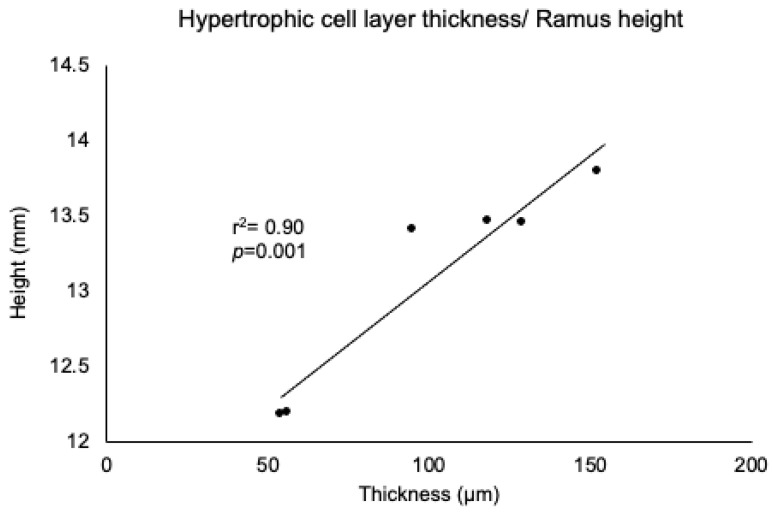
Correlation between ramus height and hypertrophic cell layer thickness. Correlations were determined between ramus height and hypertrophic cell layer thickness for each of the six groups. Abbreviations: r^2^ = squared correlation coefficient; *p* = probability.

**Figure 9 jcm-11-07359-f009:**
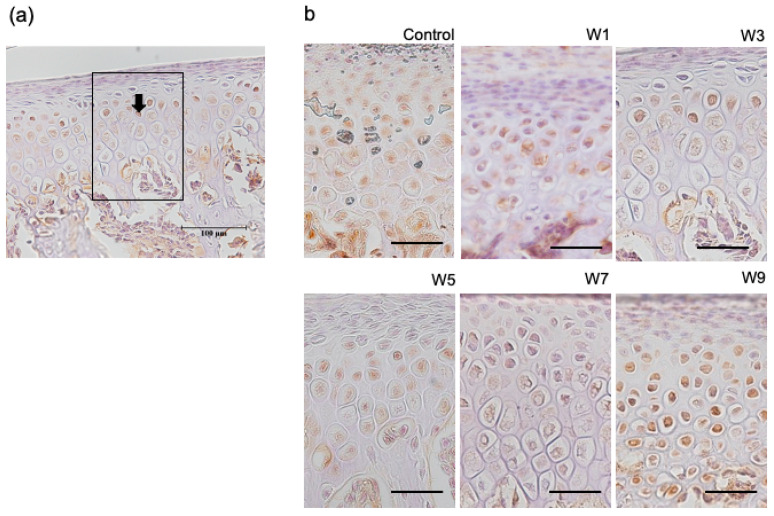
Immunohistochemical staining with anti-HIF-1α antibody. (**a**) HIF-1α-positive cells (arrows) were counted in the six groups using a fixed measuring frame (150 μm × 200 μm) indicated by the region of interest (black square). Bar depicts 100 μm; (**b**) immunohistochemically-stained images of the region of interest for the six groups. Images for HIF-1α in each of the 6 groups. Bar depicts 50 μm. Abbreviation: HIF-1α—hypoxia-inducible factor-1α.

**Figure 10 jcm-11-07359-f010:**
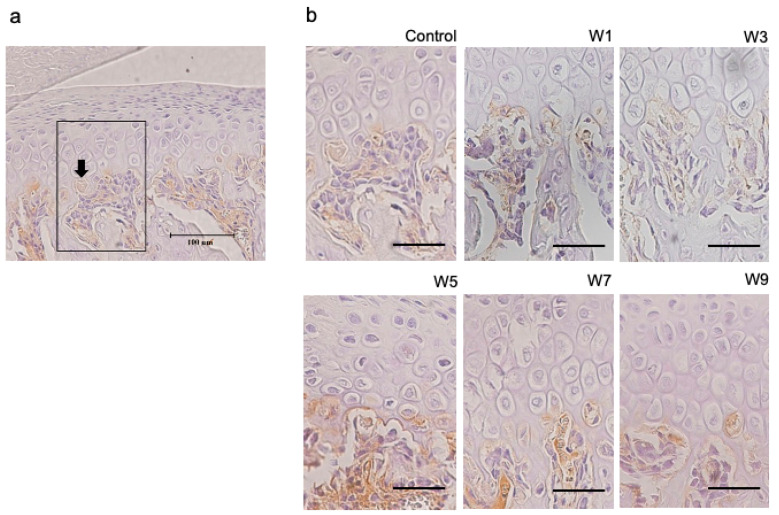
Immunohistochemical staining with RANKL. (**a**) RANKL-positive cells (arrows) were counted in the six groups using a fixed measuring frame (150 μm × 200 μm) indicated by the region of interest (black square); (**b**) immunohistochemically-stained images of the region of interest for the six groups. Images for RANKL in each of the 6 groups. Bar depicts 50 μm. Abbreviation: RANKL—receptor activator of nuclear factor kappa-B ligand.

**Figure 11 jcm-11-07359-f011:**
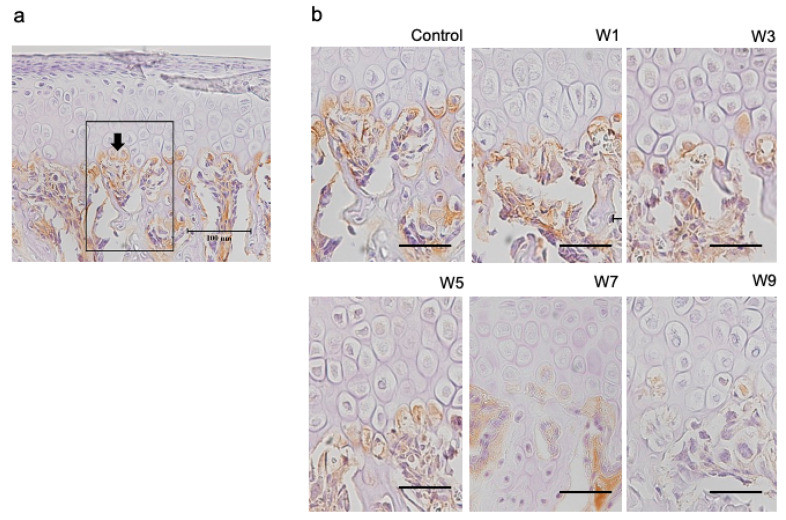
Immunohistochemical staining with OPG. (**a**) OPG-positive cells (arrows) were counted in the six groups using a fixed measuring frame (150 μm × 200 μm) indicated by the region of interest (black square); (**b**) immunohistochemically-stained images of the region of interest for the six groups. Images for OPG in each of the 6 groups. Bar depicts 50 μm. Abbreviation: OPG—osteoprotegerin.

**Figure 12 jcm-11-07359-f012:**
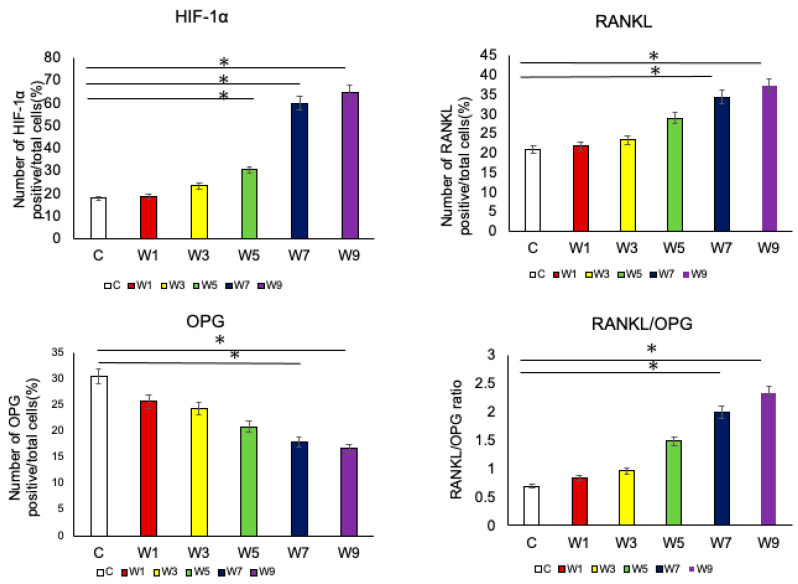
Changes in the percentages of immunohistochemically-positive cells. Comparison of the number of positive cells for antibodies against HIF-1α, RANKL, and OPG in the six groups. A comparison of the RANKLE/OPG ratios is also shown, *: *p* < 0.05. Abbreviations: HIF-1α—hypoxia-inducible factor-1α; RANKL—receptor activator of nuclear factor kappa-B ligand; OPG—osteoprotegerin.

**Table 1 jcm-11-07359-t001:** Definition of landmarks.

Landmarks	Definition
E	The intersection between the frontal bone and the most superior and anterior point of the ethmoid
U	The intersection between the maxillary sinus and the distal surface of the third superior molar tooth
Co	The most posterior and superior point on the mandibular condyle (Co, right; Co2, left)
Go	The most posterior point on the mandibular ramus
Mn	The most concave portion of the concavity on the inferior border of the mandibular corpus
Gn	The most inferior point on the ramus that lies on a perpendicular bisector of the line Go–Mn
Me	The most inferior and anterior point of the lower border of the mandible
L1	The most anterior and superior point on the alveolar bone of the mandibular incisor
Mi	The junction of the alveolar bone and the mesial surface of the first mandibular molar
ANS	The most anterior part of the palate
PNS	The most posterior part of the palate
X1	The most anterior and superior point in the molar process of the right maxilla
X2	The most anterior and superior point in the molar process of the left maxilla

**Table 2 jcm-11-07359-t002:** Measurements and interpretation.

Measurements	Interpretation
X1–X2	Maxillary width
ANS–PNS	Maxillary length
U–E	Maxillary height
Co–L1	Total mandibular length
Mi–L1	Anterior mandibular length
Go–Mn	Posterior mandibular length
Co–Gn	Ramus height
Co–Me	Mandibular body length
Co1–Co2	Mandibular width

## Data Availability

Not applicable.

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
