# Peer review of "Differential Recovery Patterns of the Maxilla and Mandible after Eliminating Nasal Obstruction in Growing Rats"

_jcm, 2022, doi:10.3390/jcm11247359_

Round 1

Reviewer 1 Report

In the study, the authors examined the maxillary-mandibular development of rats by closing one nostril and restricting nasal breathing, thus mimicking mouth breathing.

• The introduction is well written and the literature is presented in an efficient way.

• The material-methods section is explained step-by-step and expressed in clear language.

• My concern is the mouth breathing volumes of the rats whose single nostrils were closed, as stated in the limitation section. Because in the condition of adenoid hypertrophy, people are completely or almost completely mouth breathing. It is given as a limitation, thus I don't think it needs an additional correction.

Author Response

Thank you very much for providing important comments. We agree with you on the issue of the mouth breathing volumes of the rats whose unilateral nostril was closed. We would like to interpret the results in light of this and will keep this in mind for future work.

Reviewer 2 Report

The aim of this study was to clarify whether the maxillofacial morphology could attain normal growth after elimination of nasal obstruction (NO) and to determine the optimal timing of intervention for NO using an experimental model in growing rats.

The topic of the study is interesting and relevant. The paper has been written in a clear way.

In the Introduction the Authors provided some general information about nasal obstruction (NB) and its clinical implications. They also underlined the importance of the research. The aim of the study has been precisely defined.

The methodology applied has been discussed in sufficient details.

The results are quite clearly presented and the statistical analyses used in the study are adequate.

In the Discussion the Authors discussed the obtained results and compared with other findings. However, they also discussed the limitations of the study.

References should be cited according to the guidelines for authors e.g.

Journal Articles:
1. Author 1, A.B.; Author 2, C.D. Title of the article. 
Abbreviated Journal Name YearVolume, page range.

Author Response

Thank you very much for providing important comments. In response to your suggestion, we have revised the references on pages 16 and 17 in according to the guidelines. Thank you again for your kind attention and cooperation in checking it.